# Transcriptomic Profiling Reveals Distinct Immune Dysregulation in Early-Stage Sepsis Patients

**DOI:** 10.3390/ijms26146647

**Published:** 2025-07-11

**Authors:** Safa Taha, Khaled Bindayna, Muna Aljishi, Ameera Sultan, Nourah Almansour

**Affiliations:** 1Princess Al Jawhara Center for Molecular Medicine, Genetics and Inherited Diseases, Department of Molecular Medicine, College of Medicine and Health Sciences, Arabian Gulf University, Manama P.O. Box 26671, Bahrain; munajma@agu.edu.bh (M.A.); ameeraa@agu.edu.bh (A.S.); 2Department of Microbiology & Immunology, College of Medicine and Health Sciences, Arabian Gulf University, Manama P.O. Box 26671, Bahrain; bindayna@agu.edu.bh; 3Immunology and Microbiology Department, Dasman Diabetes Institute, Kuwait City 15462, Kuwait; nourah.almansour@dasmaninstitute.org

**Keywords:** sepsis, transcriptomics, differential gene expression, innate immunity, microarray, early diagnosis

## Abstract

Sepsis is a life-threatening condition characterized by dysregulated immune responses to infection. To elucidate early transcriptional changes in sepsis, we conducted a case–control study profiling gene expression in whole blood from 20 early-stage sepsis patients and 9 healthy controls. Using Affymetrix Clariom D Human Arrays and robust preprocessing, we identified differentially expressed genes (DEGs) using standard bioinformatic pipelines. A total of 344 genes were significantly upregulated, while 9703 were significantly downregulated in sepsis patients (|log2FC| > 1, adjusted *p* < 0.05). Pathway enrichment and Gene Ontology analysis revealed activation of innate immune pathways, neutrophil degranulation, and cytokine signaling, alongside suppression of lymphocyte differentiation and antigen presentation. These results suggest a shift toward an innately driven inflammatory state in early sepsis. Our findings provide transcriptomic insights that may support the development of early diagnostic biomarkers and therapeutic targets.

## 1. Introduction

Sepsis represents a complex syndrome of systemic inflammation resulting from infection and is a leading cause of mortality in intensive care units globally [1]. It is increasingly understood not as a uniform disease but as a dynamic continuum of immunological states, ranging from hyperinflammatory responses to immunosuppression [2]. This heterogeneity makes diagnosis and treatment particularly challenging. With current biomarkers like procalcitonin and CRP offering limited specificity and prognostic power [3,4,5,6,7], there is a growing demand for precision tools to characterize the molecular events that define sepsis onset and progression.

Recent advancements in transcriptomic profiling offer a window into these molecular dynamics. High-throughput gene expression analysis enables the identification of unique immune signatures, paving the way for biomarker discovery, mechanistic insights, and stratified therapy [8,9,10,11]. Notably, blood-based transcriptomic profiling serves as a minimally invasive yet informative tool that captures systemic host responses in real time [12,13]. By characterizing the transcriptional programs active during early sepsis, we can uncover key regulatory pathways involved in immune activation, metabolic reprogramming, and organ dysfunction [2,14,15,16,17,18,19,20].

Previous transcriptomic studies, such as those by [4,6], have characterized host responses in sepsis across diverse cohorts, primarily focusing on broad immune signatures or longitudinal outcomes [8,9]. In contrast, this study specifically targets early-stage sepsis within 72 h of symptom onset, capturing acute transcriptional changes in a Middle Eastern cohort using high-resolution Affymetrix Clariom D Arrays. By integrating advanced network analyses like WGCNA with traditional DEG and clustering approaches, we uncover modular gene networks that reveal novel regulatory hubs, offering unique insights into immune–metabolic reprogramming. These findings extend beyond the existing literature by identifying candidate biomarkers for early diagnosis and potential therapeutic targets tailored to the initial hyperinflammatory phase of sepsis.

In the present study, Figure 1, we used Affymetrix Clariom D Human Arrays to comprehensively profile blood gene expression in early-stage sepsis patients compared to healthy controls. Integrating differential gene expression, enrichment analyses, and machine learning clustering techniques, we identified upregulated pathways related to neutrophil activation and cytokine signaling [3,6,21], as well as the suppression of translational, metabolic, and antigen-presentation machinery [4,7,22]. These transcriptomic alterations mirror the paradoxical immune profile observed in sepsis, where exaggerated inflammation coexists with profound immune paralysis [2,23,24,25]. This expanded introduction sets the stage for a deeper investigation of the early molecular shifts that may underlie clinical outcomes in sepsis.

## 2. Results

Below is a schematic representation of the case–control study comparing early-stage sepsis patients (*n* = 20) with age- and sex-matched healthy controls (n = 9). The diagram outlines participant enrollment, consent and ethical approvals, sample collection, and downstream transcriptomic analysis. Visual elements highlight group differentiation and overall study flow.

### 2.1. Multidimensional Scaling and Differential Expression Analysis

Multidimensional Scaling (MDS) of normalized gene expression data revealed clear separation between early-stage sepsis patients and healthy controls along dimension 1, indicative of distinct global transcriptomic landscapes (Figure 2A). Healthy samples clustered tightly, reflecting high transcriptional uniformity, whereas sepsis samples exhibited greater dispersion, highlighting heterogeneous immune responses among patients, consistent with clinical variability in sepsis outcomes [8,9,26]. This clear segregation validated dataset quality and supported robust downstream analyses.

Differential expression analysis identified a total of 10,037 significantly dysregulated genes between sepsis patients and healthy controls (|log2FC| > 1, adjusted *p* < 0.05), comprising 344 upregulated and 9703 downregulated genes (Figure 2B). The volcano plot visually underscored balanced yet pronounced patterns of immune activation and suppression, reflecting a complex host response characteristic of early-stage sepsis [1,2,8].

Top significantly upregulated genes included PCDHB11 (log2FC = 1.64), MIR1185-2 (log2FC = 1.42), MIR3156-2 (log2FC = 1.41), MIR1185-1 ENSG00000221525 (log2FC = 1.38), and MIR4323 (log2FC = 1.35). These genes have been associated with regulatory roles in immune responses and inflammatory pathways, highlighting their potential involvement in the innate immune activation observed in sepsis [3,6,21,23].

In contrast, significantly downregulated genes included FBXO7 (log2FC = −5.62), EIF1AY (log2FC = −5.41), HLA-DRA (log2FC = −5.36), MKRN1 (log2FC = −5.31), and RPL10 (log2FC = −5.30). Notably, the substantial suppression of HLA-DRA, critical for antigen presentation and adaptive immune responses, highlights significant impairment in adaptive immune function, potentially contributing to immune paralysis during severe septic states [7,17,20,27]. EIF1AY and RPL10 further underscore substantial suppression of protein synthesis processes, aligning with observations from bi-clustering and enrichment analyses [22,26,28].

Functional annotation of these dysregulated genes emphasized significant suppression of pathways related to antigen presentation, oxidative homeostasis, and protein synthesis, indicating a strategic reallocation of cellular resources from adaptive immunity towards innate immune activation and energy conservation during systemic inflammatory stress [2,26,29].

The coordinates of each sample in an MDS plot and a full list of differentially expressed genes are provided in Appendix A.

Gene Set Enrichment Analysis (GSEA) confirmed these findings, highlighting significant negative enrichment (NES < −2.4, adjusted *p* < 0.0001) of key translational and biosynthetic pathways such as co-translational protein targeting to membranes, SRP-dependent co-translational protein targeting, protein targeting to the endoplasmic reticulum (ER), establishment of protein localization to endoplasmic reticulum, and cytoplasmic translation (Figure 3). Specifically, crucial genes involved in these pathways, including SEC62, SEC63, SEC61A1, RPL18, RPS20, and HSPA5, showed coordinated suppression, indicating a substantial disruption in protein synthesis and cellular maintenance mechanisms under septic conditions [2,22,30,31].

Conversely, pathways positively enriched in early sepsis prominently included innate immune activation and inflammatory signaling processes. Notable positively enriched pathways encompassed Toll-like receptor (TLR) signaling, cytokine-mediated interactions, neutrophil immunity, and NF-κB signaling, underscoring a pronounced early inflammatory response [32,33,34,35,36]. These pathways are critically involved in the initial recognition and rapid immune response to pathogens, suggesting that the transcriptional profile in early sepsis prioritizes rapid innate immune activation over cellular homeostasis and metabolism [2,8,37,38,39].

GSEA output metrics are detailed in Appendix A.

The enrichment network map shows significantly downregulated gene sets (NES < −2.5, FDR q < 0.01) in early-stage sepsis. The nodes represent individual gene sets, and the edges indicate functional overlaps based on shared genes. The clusters highlight key suppressed biological programs, including translation initiation, ribosome biogenesis, oxidative phosphorylation, and mRNA catabolism. The network underscores a coordinated shutdown of biosynthetic and metabolic pathways in response to immune activation.

### 2.2. K-Means Clustering and Functional Module Analysis

K-means clustering of the normalized expression matrix identified six transcriptional modules, labeled A through F, each representing a distinct biological program associated with early-stage sepsis (Figure 4). The number of genes within each cluster ranged from 22 in Cluster A to 658 in Cluster D, highlighting variable modular complexity.

Among these, Cluster B (71 genes) showed robust enrichment for immune-related processes, particularly those involving neutrophil activity and innate immune defense. Key genes in this module included SPI1, MMP9, MMP8, S100A9, S100A12, DEFA1, and DEFA1B, all of which play central roles in neutrophil-mediated inflammation, pathogen clearance, and cytokine signaling [3,6,23]. Gene Ontology (GO) enrichment analysis for Cluster B revealed strong associations with the following:

Neutrophil activation involved in immune response (adjusted *p* = 8.2 × 10^−8^)Granulocyte degranulation (adjusted *p* = 8.2 × 10^−8^);Defense response to fungus (adjusted *p* = 4.9 × 10^−8^).

Due to the imbalance in differentially expressed genes (with the majority being downregulated), most clusters show reduced expression levels; however, cluster F is distinct in exhibiting a markedly stronger downregulation. Cluster F (290 genes) was distinguished by the widespread transcriptional suppression of genes involved in core cellular functions such as protein biosynthesis and RNA metabolism. This includes downregulation of components of the translation machinery, mRNA catabolism, and viral gene expression pathways. These signatures were consistent with results from bi-clustering and Gene Set Enrichment Analysis (GSEA), further underscoring extensive translational arrest and metabolic reprogramming during early sepsis [26,30,40,41,42]. Enriched GO terms for Cluster F included the following:

mRNA catabolic processes (adjusted *p* < 1 × 10^−50^);

Translation initiation;

Viral gene expression and antigen processing [7,43,44,45].

These findings underscore a pronounced transcriptional dichotomy during early sepsis, where acute immune activation coexists with suppression of essential biosynthetic and metabolic pathways. This suggests a strategic reallocation of leukocyte resources to support immediate immune defense, potentially at the expense of long-term cellular maintenance. Such immune–metabolic tradeoffs may contribute to the development of immunosuppression and increased susceptibility to secondary infections commonly observed in sepsis patients [2,8,17].

Details of K-means clustering and Functional Module Analysis are provided in Appendix A.

### 2.3. Bi-Clustering Analysis

Bi-clustering analysis identified a dominant co-expression module consisting of approximately 1000 genes that exhibited coordinated upregulation in healthy controls and significant suppression in sepsis patient samples (Figure 5). This substantial bi-cluster represents a critical transcriptional signature reflective of systemic transcriptional reprogramming commonly observed in early-stage sepsis.

Detailed functional annotation and literature-supported analysis revealed that genes within this bi-cluster were profoundly enriched in pathways related to ribosomal function, oxidative phosphorylation, and core metabolic processes. Key genes involved in oxidative phosphorylation, including ATP5A1, ATP5B, and ATP5J, and numerous ribosomal protein genes (e.g., RPS20, RPL31), demonstrated consistent downregulation in sepsis samples, supporting the notion of translational and metabolic suppression as a response to acute systemic inflammation [22,46,47,48,49,50].

Additional suppressed genes within the bi-cluster were identified in pathways critical for protein synthesis and cellular maintenance, including EIF4B involved in translation initiation, VAMP3 associated with vesicle-mediated transport, and DCUN1D1 involved in protein ubiquitination. The uniform suppression of these genes highlights extensive cellular reprogramming aiming to conserve energy and limit protein production during severe inflammatory stress, a phenomenon documented previously in various systemic inflammatory states, including sepsis [35,38,39,44,46,49].

Moreover, the observed downregulation of genes involved in maintaining cellular oxidative balance, such as CYB5R4, highlights impaired antioxidant defenses in septic patients, possibly exacerbating oxidative damage and contributing to cellular dysfunction commonly noted in sepsis-induced organ injury [17,48].

Collectively, this bi-cluster analysis underscores a coordinated transcriptional response strategy characterized by metabolic shutdown, decreased translational activity, and impaired oxidative balance. These processes represent key pathophysiological mechanisms contributing to immune dysregulation and organ dysfunction in early-stage sepsis, offering insights into potential therapeutic targets aimed at restoring cellular metabolism and function [51,52,53,54]. A list of 1000 genes identified using bi-clustering analysis is provided in Appendix A.

The heatmap shows a dominant co-expression bi-cluster of approximately 1000 genes across 29 samples, including healthy controls and early-stage sepsis patients. The expression matrix is z-score normalized per gene (row-wise), with red indicating high expression, green low expression, and black representing intermediate expression. Genes in this bi-cluster show consistent upregulation in healthy controls and suppression in sepsis patients, capturing a robust transcriptional signature associated with translational arrest and metabolic downregulation.

### 2.4. WGCNA Identifies Distinct Co-Expression Modules Altered in Sepsis

To explore higher-order transcriptional relationships, we applied Weighted Gene Co-expression Network Analysis (WGCNA) to high-variance genes [49]. This analysis identified several distinct gene modules significantly associated with the sepsis phenotype, notably the turquoise and blue modules (Figure 6).

The turquoise module (Module 1), comprising 835 genes, exhibited a strong positive correlation with sepsis status (r = 0.76, *p* < 0.01). Genes in this module, such as 15-SEP, 7-MAR, 7-SEP, and AC011404.1, were enriched in biological processes related to neutrophil activation, cytokine signaling, and Toll-like receptor pathways—aligning closely with differential gene expression (DEG) and enrichment analysis findings [24,37].

In contrast, the blue module (Module 2), containing 164 genes, was negatively correlated with sepsis (r = –0.68). Representative genes within this module included AC013394.2, AP000240.9, and CTD-2370N5.3, which were primarily involved in ribosome biogenesis, oxidative phosphorylation, and protein synthesis pathways [23,41]. This aligns with the observed suppression patterns noted in the bi-clustering and Gene Set Enrichment Analysis (GSEA) [26].

Functional annotation of these modules confirmed significant immune–metabolic reprogramming and translational suppression during early sepsis, with WGCNA highlighting modular gene networks that underpin and extend insights derived from traditional DEG analyses [42,45,48]. A full list of genes with their designated clusters, identified using Weighted Gene Co-expression Network Analysis (WGCNA), is provided in Appendix A.

## 3. Discussion

Our comprehensive transcriptomic analysis of early-stage sepsis reveals coordinated and multifaceted immune dysregulation characterized by simultaneous upregulation of innate immune responses and downregulation of adaptive immunity, antigen presentation, protein synthesis, and metabolic homeostasis. These findings reinforce the dual-phase paradigm of sepsis, where an early hyperinflammatory response coexists with cellular exhaustion and immunosuppression [1,2,10].

Multidimensional Scaling (MDS) demonstrated clear separation between septic patients and healthy controls, highlighting significant global transcriptomic shifts. Increased dispersion among sepsis samples reflects heterogeneous immune responses, consistent with clinical variability in disease progression and outcomes [10,11,30].

Differential expression analysis identified significant upregulation of genes involved in neutrophil activation and inflammatory signaling, including *OLFM4*, *LCN2*, and *S100A12*. These genes play roles in antimicrobial defense, tissue infiltration, and cytokine-mediated feedback loops, exacerbating tissue damage during sepsis [3,4,5,24]. These transcriptional signatures align closely with pathways governed by Toll-like receptor (TLR) signaling and NF-κB activation, emphasizing innate immune activation dominance [24,37].

Conversely, markedly downregulated transcripts included *HLA-DRA*, *CTSS*, and TXNIP, along with broad sets of ribosomal and oxidative phosphorylation genes. Gene Set Enrichment Analysis (GSEA) confirmed suppression of key biosynthetic processes, including ribosome biogenesis, translation initiation, mRNA catabolism, and oxidative metabolism [23,26,34,41]. These shifts indicate immune-metabolic reprogramming of leukocytes, prioritizing inflammatory responses over cellular maintenance functions—potentially serving as an energy conservation mechanism during systemic stress [42,46,47].

K-means clustering and bi-clustering analyses provided complementary insights into the transcriptional landscape of early-stage sepsis. K-means clustering revealed distinct gene expression modules, including Cluster B, enriched for neutrophil-mediated immune responses, and Cluster F, characterized by widespread downregulation of genes involved in translation and protein synthesis. These modules reflect the dual nature of sepsis pathophysiology, where hyperactivation of innate immunity coexists with suppression of cellular maintenance programs.

In parallel, bi-clustering analysis uncovered a dominant transcriptional signature comprising approximately 1000 genes that were consistently highly expressed in healthy controls but markedly suppressed in sepsis patients. This bi-cluster encompassed key components of ribosomal function, oxidative phosphorylation, and mRNA processing. Notably, ATP synthase subunits (*ATP5A1*, *ATP5B*), cytoskeletal proteins such as VIM (Vimentin), and numerous ribosomal protein genes (*RPS20*, *RPL31*) exhibited profound transcriptional repression [23,41].

The consistent downregulation of these core homeostatic and metabolic pathways underscores a global transcriptional reprogramming aimed at reallocating cellular energy toward immune defense rather than biosynthetic activities. While this strategy may enable rapid inflammatory responses, it comes at the cost of translational arrest and metabolic shutdown, potentially contributing to immune exhaustion, impaired tissue repair, and organ dysfunction. These findings highlight critical vulnerabilities in septic leukocytes that may serve as targets for therapeutic intervention, particularly strategies aimed at restoring metabolic balance and protein synthesis capacity during recovery from sepsis [42,44].

Extending these findings, Weighted Gene Co-expression Network Analysis (WGCNA) uncovered additional regulatory architecture by clustering genes into co-expressed modules with distinct biological functions correlated to disease status [49]. The turquoise module, highly associated with sepsis, was enriched with genes involved in neutrophil-mediated immunity, cytokine signaling, and TLR pathways [24,37]. Conversely, negatively correlated blue and green modules contained genes responsible for ribosome structure, translation, mitochondrial respiration, and antigen presentation, mirroring the suppressed pathways observed in GSEA and bi-clustering [23,26,41]. These insights underscore that immune and metabolic systems are regulated through highly coordinated modular networks, extending beyond individual gene activation.

WGCNA enhances transcriptomic profiling resolution by identifying functional units beyond DEGs, reinforcing the reproducibility of the immune-metabolic switch observed in early sepsis [42,45,48]. The convergence of MDS, DEG, GSEA, clustering, and WGCNA analyses robustly supports the dual-phase model of sepsis pathogenesis [10,22,42].

The identified transcriptomic signatures, particularly the upregulation of neutrophil-associated genes (*OLFM4*, *S100A12*) and the suppression of adaptive immunity markers (HLA-DRA), hold significant potential for clinical translation. Unlike existing biomarkers like procalcitonin or CRP, which lack specificity for early sepsis [8,9], our gene signatures could enable precise risk stratification and early diagnosis within the critical first 72 h. For instance, a diagnostic panel targeting *OLFM4* and *S100A12* could identify patients at risk of rapid disease progression, guiding timely interventions such as antibiotics or immunomodulatory therapies. Additionally, the suppression of translational and metabolic pathways (e.g., *ribosomal genes*, *ATP5A1*) suggests therapeutic opportunities to restore cellular homeostasis, potentially via metabolic modulators or immunotherapies targeting immune exhaustion. These biomarkers could be integrated into point-of-care assays or machine learning models to personalize sepsis management.

These modules offer practical translational potential, containing candidate hub genes that could serve as early biomarkers or therapeutic targets for stratifying sepsis patients and modulating inflammatory-metabolic balances. Future studies should validate these signatures across broader longitudinal patient cohorts to assess their diagnostic and prognostic efficacy [40,45,48].

Together, our findings illustrate a complex transcriptional landscape in early-stage sepsis, highlighting immune escalation and suppression of cellular homeostasis. Integrative network-based methods like WGCNA offer new interpretative avenues for immune dynamics, identifying actionable gene regulatory nodes.

A limitation of this study is the reliance on microarray-based transcriptomic profiling without orthogonal validation of key differentially expressed genes (DEGs) such as *HLA-DRA*, *OLFM4*, and *S100A12*. While microarray data were rigorously quality-controlled and normalized, future studies should employ quantitative real-time PCR (qRT-PCR) to confirm these transcriptional changes. Additionally, protein-level validation using techniques such as ELISA or flow cytometry for *HLA-DR* and LCN2 could further substantiate the translational relevance of these findings, bridging the gap between transcriptomic signatures and clinical applicability. Such validations are planned as part of our ongoing research to strengthen the identified biomarkers’ diagnostic and prognostic potential.

## 4. Materials and Methods

### 4.1. Study Design

This study utilized a case–control design to investigate gene expression patterns in early-stage sepsis. We included two groups of adult participants aged 18 years and older: Group 1 comprised 10 patients diagnosed with early-stage sepsis, while Group 2 included 9 healthy controls. Patients were recruited from the Intensive Care Unit (ICU) at Salmaniya Medical Complex (SMC) in Bahrain. Healthy controls, free from infections or inflammatory conditions, were selected from blood donors at the SMC Blood Bank, ensuring that age and sex were matched between the groups. All participants provided informed written consent, and ethical approvals were obtained from the College of Medicine and Health Sciences at Arabian Gulf University, SMC, and the Ministry of Health in Bahrain.

### 4.2. Early-Stage Sepsis

The clinical diagnosis of sepsis was based on the Sepsis-3 definitions, which require either suspected or confirmed infection alongside an increase in the Sequential Organ Failure Assessment (SOFA) score of 2 or more points. This diagnosis considers symptoms onset within the last 72 h. Patients must exhibit at least one of the following abnormal vital signs:

Fever (≥38.0 °C) or hypothermia (≤36.0 °C), tachycardia (heart rate > 90 beats per minute), tachypnea (respiratory rate >20 breaths per minute), or altered mental status.

Septic Shock: A subset of sepsis characterized by persistent hypotension requiring vasopressors to maintain a mean arterial pressure (MAP) ≥ 65 mmHg and a lactate level >2 mmol/L despite adequate fluid resuscitation.

### 4.3. Healthy Controls

Controls were individuals without any history of infection or sepsis, matched for age and gender with the patient group. They had no chronic illnesses that could influence immune response, such as diabetes or autoimmune diseases.

### 4.4. Exclusion Criteria

Patients were excluded if they had undergone major surgery within the last 30 days, were immunocompromised (e.g., receiving chemotherapy or corticosteroids), were pregnant, or had concurrent infections complicating the diagnosis of sepsis. Additionally, individuals under 18 years of age were excluded (Table 1).

### 4.5. Blood Sample Collection and RNA Extraction

Blood samples were collected using Tempus^TM^ Blood RNA Tubes (Applied Biosystems^TM^, Waltham, MA, USA) and vortexed for 10 s to ensure thorough mixing with the stabilizing reagent. RNA extraction was performed using the Stabilized Blood-to-CT^TM^ Nucleic Acid Preparation Kit, following the manufacturer’s guidelines. Initially, 500 μL of each Tempus-stabilized blood sample was transferred to a 1.5 mL microcentrifuge tube. To this, 250 μL of 2X PBS was added, followed by vortexing for 10 s and brief centrifugation. Next, 20 μL of Tempus^TM^ Pellet Enhancer was added; the mixture was vortexed again for 10 s and centrifuged at 5000× *g* for 10 min. The supernatant was carefully removed, leaving approximately 50 μL behind with the pellet. Washing steps were repeated three to five times: first, 750 μL of Tempus^TM^ (Chicago, IL, USA) Washing Buffer 1 was added and vortexed until the pellet dissolved, then centrifuged at 5000× *g* for 2 min. The washing buffer was removed, retaining about 20 μL of residual supernatant. Subsequently, 750 μL of Tempus^TM^ Washing Buffer 2 was added, vortexed briefly, and centrifuged again at 5000× *g* for 2 min, after which the tube with the pellet was placed on ice. To eliminate any genomic DNA, a digestion solution mixed with DNase I (1:100) was prepared and added to a microcentrifuge tube. After mixing, 129 μL of this digestion solution was added to the pellet, which was pipetted up and down five times to break it up. The mixture was incubated at room temperature for 8 min. Finally, 10 μL of Stop Solution was added, and the sample was incubated for an additional 2 min before being stored at –80 °C for further analysis.

### 4.6. Microarray Data Generation and Quality Control

Gene expression profiling was performed using the Affymetrix Clariom D Human Array. The quality of the extracted RNA from peripheral blood mononuclear cells (PBMCs) was assessed using an Agilent Bioanalyzer. The purified RNA was used as a template for first strand cDNA synthesis following the “GeneChip^®^ WT PLUS Reagent Kit” (Thermo Fisher Scientific, Santa Clara, CA, USA). The first amplification was carried out from a total of 250 ng of initial RNA. Subsequently, the second strand of cDNA was synthesized. This double-stranded cDNA was then used to synthesize cRNA, which was purified, and its concentration measured. A second cycle of cDNA synthesis was performed, followed by hydrolysis, purification, fragmentation, and labeling. Finally, the labeled fragmented cDNA was hybridized to the GeneChip^®^ Clariom D Human Array (Thermo Fisher Scientific) at 45 °C for 16 h in the Affymetrix GeneChip^®^ Hybridization Oven 645. After hybridization, the chips were washed and stained using the Affymetrix GeneChip^®^ Fluidics Station 450, following the procedures described in the “GeneChip^®^ WT PLUS wash and stain Kit” protocol. All chips were scanned at room temperature using the Affymetrix GeneChip^®^ Scanner 3000 7G. Data preprocessing steps included importing CEL files into the Transcriptomic Analysis Console (TAC) software. The Transcriptome Analysis Console (TAC) software version 4.0.0.25 by Thermo Fisher Scientific Quality control methods involved visual inspections of boxplots and density plots to assess data distribution and quality. Normalization of the raw expression data was performed using the Robust Multi-array Average (RMA) method, which includes background correction, quantile normalization, and summarization of probe sets. All analyses were performed using R (version 4.4.1) and Bioconductor (version 3.19). The Transcriptomic Analysis Console (TAC, version 4.0.2, Thermo Fisher Scientific) was used for data preprocessing. The ‘affy’ package (version 1.78.0) implemented the Robust Multi-array Average (RMA) method for normalization.

Principal Component and Clustering Analysis: Principal Component Analysis (PCA), Multidimensional Scaling (MDS), and t-distributed Stochastic Neighbor Embedding (t-SNE) were applied to RMA-normalized gene expression data using the Transcriptomic Analysis Console (TAC). MDS revealed that dimension1 accounted for the greatest variance and clearly separated sepsis patients from healthy controls. Additional components (PC2–PC5) supported sample-specific variance. MDS and t-SNE analyses confirmed the distinct transcriptomic profiles of patient and control groups.

Differential Expression Analysis: DEGs between sepsis patients and healthy controls were identified using linear modeling and empirical Bayes statistics. Genes with |log2 fold change| > 1 and adjusted *p*-value < 0.05 were considered significant. A volcano plot was generated to visualize global transcriptional changes, plotting log2 fold change against -log10-adjusted *p*-value.

Pathway and Functional Enrichment: Enrichment analyses were performed using DAVID and Reactome databases. Gene Ontology (GO) terms and KEGG pathways were used to interpret biological significance.

K-Means Clustering and Functional Module Analysis: To identify gene co-expression modules, unsupervised K-means clustering was applied to the normalized expression matrix. K-means clustering was performed using the ‘stats’ package (version 4.4.1) with k = 6, selected based on the elbow method to optimize within-cluster sum of squares, ensuring biologically meaningful gene modules. The ‘cluster’ package (version 2.1.6) was used for silhouette analysis to validate cluster quality. Genes were grouped into six major clusters (A–F), which were then visualized via heatmap. Functional enrichment analysis was performed for each cluster using GO biological processes with adjusted *p*-values.

Bi-clustering Analysis: To identify coherent gene modules regulated across subsets of samples, we performed bi-clustering using the biclust R package. Unlike traditional clustering, which captures patterns across all samples, bi-clustering identifies genes that are co-expressed within specific sample subsets. We applied the BCPlaid algorithm under default parameters, modeling the expression matrix as a sum of additive layers to capture distinct transcriptional patterns. This approach enabled detection of a large co-expression bi-cluster associated with coordinated downregulation of metabolic and translational programs in sepsis patients. Further details are provided in Appendix A.

Co-expression Network Analysis: To identify co-regulated gene modules associated with early-stage sepsis, we implemented Weighted Gene Co-expression Network Analysis (WGCNA) using R. WGCNA was implemented using the ‘WGCNA’ package (version 1.72-5). A soft-thresholding power of β = 6 was chosen based on scale-free topology fit (R^2^ > 0.85), determined using the pickSoftThreshold function. The minimum module size was set to 30 genes, and dynamic tree cutting was performed with a deepSplit of 2 to ensure robust module detection.

A practical clinical scenario for applying these transcriptomic signatures involves the rapid identification of early-stage sepsis in emergency departments. For example, a patient presenting with fever and tachycardia could undergo a blood-based transcriptomic assay targeting OLFM4, S100A12, and HLA-DRA. Elevated OLFM4 and S100A12 combined with suppressed HLA-DRA could confirm sepsis within hours, enabling the prompt initiation of antibiotics and fluids, potentially reducing progression to septic shock. Such an assay, deployable via point-of-care platforms, could integrate with existing electronic health record systems to stratify patients by risk of organ dysfunction (e.g., high SOFA score) and guide immunomodulatory therapies, such as anti-inflammatory agents for hyperinflammatory profiles or immune stimulants for immunosuppressive states [42,45]. This approach addresses the current diagnostic delay and heterogeneity in sepsis management.

Normalized gene expression data were filtered to include genes with sufficient variance across samples. An unsigned adjacency matrix was created using a soft-thresholding power β, selected based on scale-free topology criteria. The topological overlap matrix (TOM) was computed to measure network connectivity. Genes were then hierarchically clustered and grouped into modules using dynamic tree cutting. Module eigengenes (MEs) were calculated to represent the expression profile of each module, and their correlation with clinical traits (sepsis vs. control) was assessed. Modules significantly correlated with sepsis status underwent functional enrichment analysis using GO terms and Reactome pathways.

## 5. Conclusions

This study provides a comprehensive transcriptomic landscape of early-stage sepsis, illustrating dual-pattern immune dysregulation characterized by innate immune hyperactivation and concurrent suppression of adaptive immunity, translation, and metabolism. Employing an integrative systems biology approach—MDS, differential expression, pathway enrichment, GSEA, K-means clustering, bi-clustering, and WGCNA—we identified multi-scale transcriptional signatures distinguishing sepsis patients from healthy controls.

Upregulated genes and pathways underscored excessive neutrophil activation, inflammatory signaling, and innate immune priming, whereas downregulated programs revealed widespread impairments in antigen presentation, protein synthesis, and energy metabolism. These concurrent opposing shifts highlight the immune paradox of sepsis: simultaneous hyperinflammation and immune paralysis [1,2,42].

Critically, WGCNA delineated distinct co-expression modules, such as the turquoise module enriched in inflammatory mediators and the blue module representing suppressed ribosomal and metabolic functions. These modular patterns validate and extend DEG findings by revealing higher-order transcriptional architecture and potential hub regulators [23,41,42,49].

Identifying these network-based modules advances current sepsis models by providing not only individual biomarkers but also functional gene circuits, informing future diagnostic or therapeutic strategies. Specifically, the suppression of translation-related and metabolic modules presents a promising axis for evaluating immune exhaustion and host resilience [42,44,45].

Collectively, this work deepens our understanding of early sepsis biology, laying the foundation for personalized, network-informed interventions targeting immune and metabolic dysregulation at the transcriptomic level.

By pinpointing actionable gene networks, our findings pave the way for point-of-care diagnostic tools and personalized therapies, potentially transforming sepsis management in critical care settings.

## Figures and Tables

**Figure 1 ijms-26-06647-f001:**
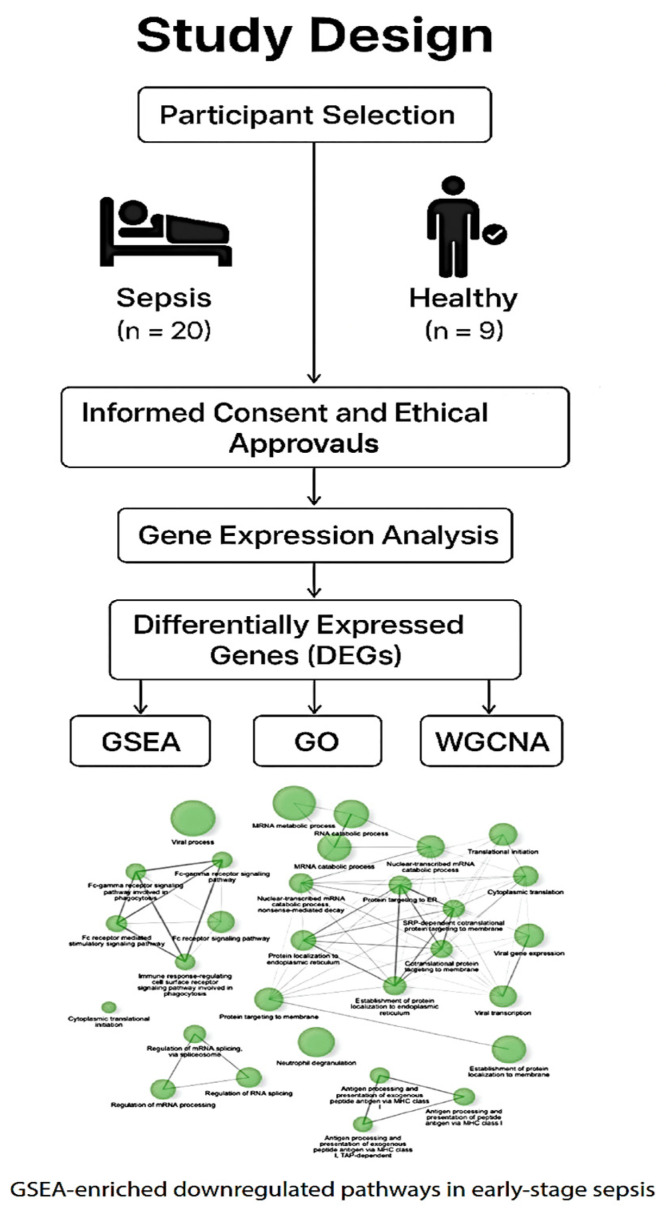
Study Design and Gene Expression Analysis Workflow This diagram outlines the study design and the workflow for gene expression analysis. The study involved the selection of participants, including 20 individuals with sepsis and 9 healthy controls. All participants provided informed consent, and the study received ethical approval. The gene expression analysis was conducted to identify differentially expressed genes (DEGs). Subsequent analyses included Gene Set Enrichment Analysis (GSEA), Gene Ontology (GO) enrichment, Weighted Gene Co-expression Network Analysis (WGCNA), and an examination of metabolic processes. The diagram also highlights the regulation of miRNA processing and the GSEA-enriched downregulated pathways in early-stage sepsis.

**Figure 2 ijms-26-06647-f002:**
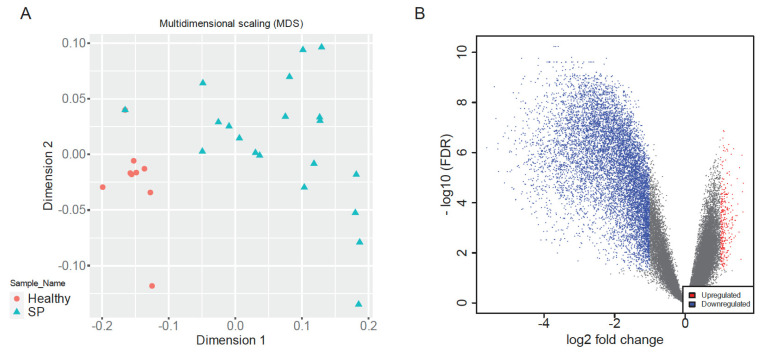
Transcriptomic separation and differential expression in early-stage sepsis. (**A**) MDS plot showing clear transcriptomic separation between sepsis patients and healthy controls along dimension 1. (**B**) The Volcano plot shows the log2 fold change (x-axis) versus the -log10(FDR) (y-axis) for each gene. Genes that are significantly upregulated are marked in red, while those that are significantly downregulated are marked in blue. The gray points represent genes that do not meet the significance threshold for differential expression. This visualization helps to identify genes with distinct immune signatures, highlighting their roles in the biological processes under study.

**Figure 3 ijms-26-06647-f003:**
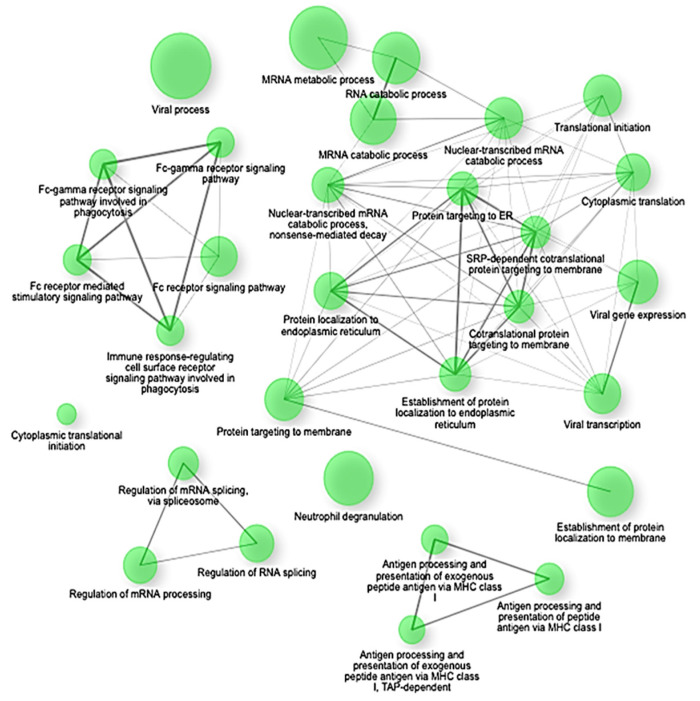
Network View of GSEA-Enriched Downregulated Pathways in Early-Stage Sepsis. This network visualization highlights the downregulated pathways in early-stage sepsis, including mRNA metabolism, RNA catabolism, viral processes, and immune response-related pathways. Nodes represent specific pathways, and edges indicate interactions between them.

**Figure 4 ijms-26-06647-f004:**
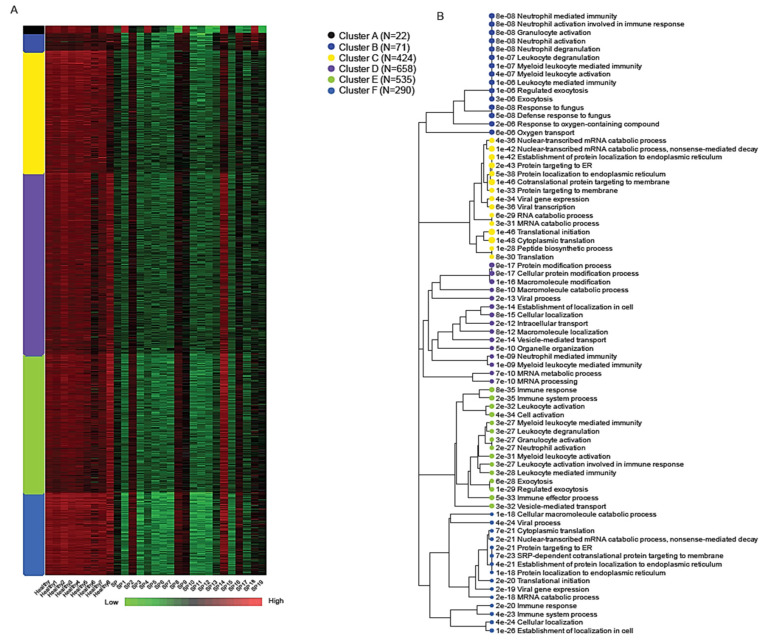
Gene clustering and functional enrichment of early sepsis transcriptome. (**A**) A heatmap showing six k-means gene expression clusters (A–F) across healthy controls and sepsis patients. Cluster B is enriched for neutrophil-mediated immune activation, while Cluster F shows the coordinated downregulation of translation-associated genes. The heatmap is row-normalized, and a scale bar indicates low (green), intermediate (black), and high (red) gene expression levels. (**B**) An enrichment tree map of biological processes associated with the six clusters. Each branch represents a GO term, with node proximity and hierarchy reflecting functional similarity. Prominent enriched processes include neutrophil activation, mRNA catabolism, vesicle-mediated transport, and translational initiation.

**Figure 5 ijms-26-06647-f005:**
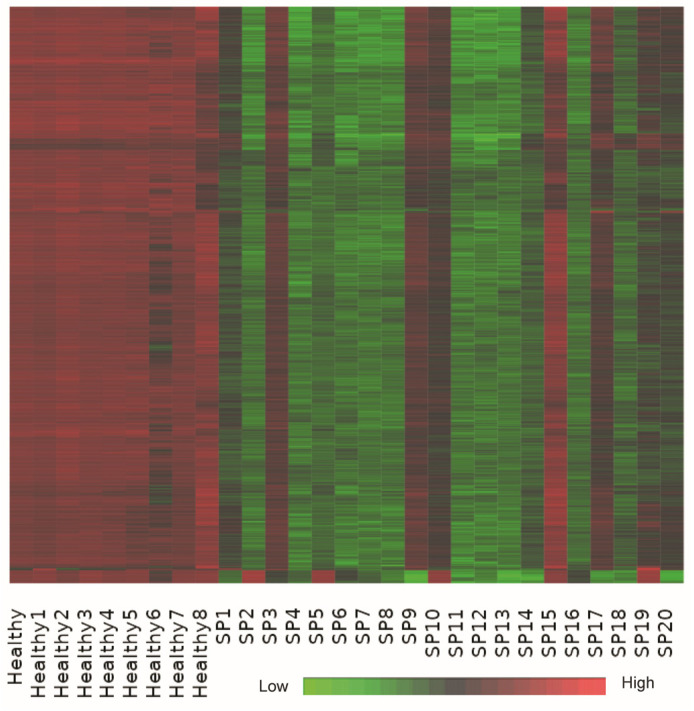
Bi-clustering reveals transcriptional suppression of metabolic and translational programs in early sepsis.

**Figure 6 ijms-26-06647-f006:**
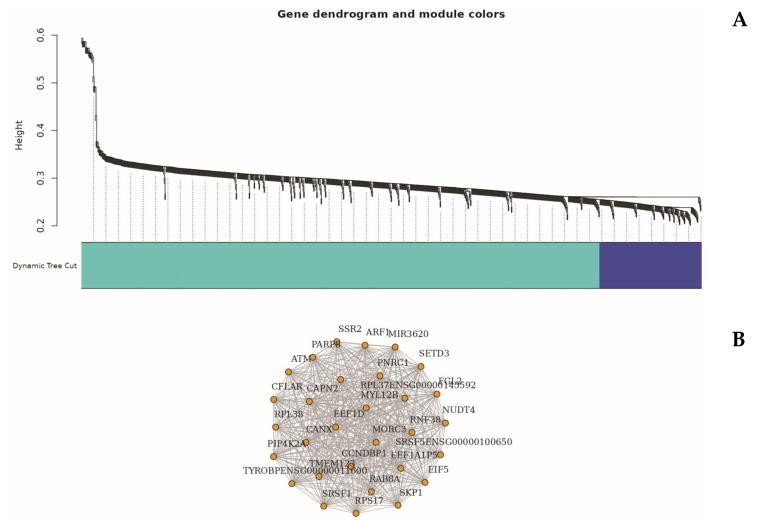
WGCNA-based co-expression modules and gene connectivity in early-stage sepsis. (**A**) Gene dendrogram with dynamic tree cut illustrating module detection by hierarchical clustering. Major modules (e.g., turquoise, blue) represent co-expressed gene sets. (**B**) Network plot of top hub genes from turquoise module within key module. Nodes represent genes, and edges denote co-expression relationships, highlighting tightly interconnected functional sub-networks.

**Table 1 ijms-26-06647-t001:** Inclusion and exclusion criteria.

Criteria	Early-Stage Sepsis	Healthy Controls
Inclusion Criteria	Clinical diagnosis of sepsis (Sepsis-3)	No history of infection or sepsis
	Symptoms onset within the last 72 h	Age and gender matched with patients
	At least one abnormal vital sign	No chronic illness affecting immune response
Exclusion Criteria	Recent major surgery within the last 30 days	Not applicable
	Immunocompromised status	
	Pregnancy	
	Concurrent infectiona complicting diagnosis	
	Individuals under 18 years of age	

## Data Availability

Data will be available upon request. To clarify, normalized gene expression matrices and de-identified clinical metadata (e.g., age, sex, sepsis status) are available upon reasonable request to the corresponding author, subject to ethics committee approval.

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
