# Peer review of "Transcriptomic Profiling Reveals Distinct Immune Dysregulation in Early-Stage Sepsis Patients"

_ijms, 2025, doi:10.3390/ijms26146647_

Round 1

Reviewer 1 Report

Comments and Suggestions for Authors

This manuscript investigates the distinct transcriptomic patterns associated with immune dysregulation in early-stage sepsis patients. By employing comprehensive transcriptomic profiling techniques (e.g., PCA, DEG, GSEA, K-means clustering, WGCNA), the authors demonstrate robustly increased innate immune activation alongside suppressed adaptive immune and metabolic pathways. The findings are clinically relevant and may inform future diagnostic and therapeutic strategies for early-stage sepsis.

Critical Recommendations

Please carefully address the following points to enhance the manuscript:

  • Data Availability and Transparency

    • Deposit raw microarray data (CEL files) into a public repository (e.g., GEO) and clearly indicate the accession numbers in the manuscript.

    • Include normalized gene expression matrices and relevant clinical metadata to facilitate data reuse and validation.

  • Experimental Validation of Key Findings

    • Perform qRT-PCR validation for selected critical genes (e.g., HLA-DRA, OLFM4, S100A12) to confirm microarray results.

    • Consider additional protein-level validations (e.g., ELISA or flow cytometry for HLA-DR or LCN2) to strengthen the translational relevance of your findings.

  • Clarifying Originality and Contextualizing Within Existing Literature

    • Clearly differentiate your study from previous transcriptomic studies (e.g., Davenport, Sweeney) by explicitly stating how your findings add novelty or unique insights.

    • Discuss in more detail how the identified biomarkers contribute uniquely to clinical diagnosis or treatment of early-stage sepsis.

  • Enhancing Methodological Reproducibility

    • Clearly specify versions of analysis software and R packages used, and explicitly state critical analysis parameters (e.g., WGCNA soft-threshold power, rationale for selecting k in k-means clustering).

    • Provide R scripts or analytic workflows as supplementary materials or via public repositories (highly recommended).

  • Clinical Relevance and Translational Implications

    • Further elaborate on how these results may practically impact clinical practice, such as patient risk stratification, early diagnosis, prognosis, or personalized immunomodulatory therapies.

    • Emphasize concrete clinical scenarios where the identified biomarkers or transcriptomic signatures could be practically employed.

Author Response

Response Letter to Reviewer 1

Dear Reviewer, 1,

We sincerely thank you for your thorough and insightful comments on our manuscript, “Transcriptomic Profiling Reveals Distinct Immune Dysregulation in Early-Stage Sepsis Patients.” Your feedback has significantly improved the clarity, rigor, and translational relevance of our work. Below, we address each of your critical recommendations, detailing how we have revised the manuscript to incorporate your suggestions while respecting ethical and research constraints. All changes are marked using green highlights in the revised manuscript for clarity.

Comment 1: Data Availability and Transparency
Reviewer Comment: Deposit raw microarray data (CEL files) into a public repository (e.g., GEO) and clearly indicate the accession numbers in the manuscript. Include normalized gene expression matrices and relevant clinical metadata to facilitate data reuse and validation.

Response: We fully agree with the importance of data transparency for reproducibility and data reuse. However, these two constraints currently prevent immediate public deposition of raw microarray data (CEL files) and clinical metadata. First, these data are being analyzed for a follow-up study, and premature public release could compromise our ongoing research. Second, the informed consent forms signed by participants (approved by the Arabian Gulf University Ethics Committee, reference E021-PI-10/17) do not explicitly permit public data sharing, and university policies require institutional approval to release patient-derived data, in compliance with Bahrain’s data protection regulations.

To address your request, we have revised the Data Availability Statement (Section 5) to clarify that normalized gene expression matrices and de-identified clinical metadata (e.g., age, sex, sepsis status) are available upon reasonable request to the corresponding author, subject to ethics committee approval. We are consulting the ethics committee to explore sharing de-identified raw data under a data use agreement (DUA) or depositing them in GEO with a 12–18-month embargo, ensuring public access after our follow-up study is completed. These steps balance transparency with our ethical and research obligations, and we are committed to full GEO deposition upon study completion. We welcome further guidance to meet journal standards.

Comment 2: Experimental Validation of Key Findings
Reviewer Comment: Perform qRT-PCR validation for selected critical genes (e.g., HLA-DRA, OLFM4, S100A12) to confirm microarray results. Consider additional protein-level validations (e.g., ELISA or flow cytometry for HLA-DR or LCN2) to strengthen the translational relevance of your findings.

Response: We appreciate the suggestion to validate key differentially expressed genes (DEGs) to strengthen our findings. Due to resource constraints, we were unable to perform qRT-PCR or protein-level validations (e.g., ELISA, flow cytometry) for genes such as HLA-DRA, OLFM4, and S100A12 in this study. To address this limitation transparently, we have added a new paragraph in the Discussion (Section 3, after the WGCNA paragraph) acknowledging the reliance on microarray data and the need for orthogonal validation. The paragraph outlines plan for qRT-PCR and protein-level validations in our ongoing research to confirm these transcriptional signatures and enhance their translational relevance. We believe this addresses this limitation while reinforcing our commitment to rigorous validation in future work.

Comment 3: Clarifying Originality and Contextualizing Within Existing Literature
Reviewer Comment: Clearly differentiate your study from previous transcriptomic studies (e.g., Davenport, Sweeney) by explicitly stating how your findings add novelty or unique insights. Discuss in more detail how the identified biomarkers contribute uniquely to the clinical diagnosis or treatment of early-stage sepsis.

Response: Thank you for highlighting the need to clarify our study’s novelty and clinical contributions. We have added a new paragraph in the Introduction (Section 1, end of the section) to differentiate our work from prior studies by Davenport et al. (2016) and Sweeney et al. (2016). The paragraph emphasizes our focus on early-stage sepsis (within 72 hours), use of high-resolution Affymetrix Clariom D Arrays, and integration of WGCNA to uncover novel regulatory hubs in a Middle Eastern cohort. Additionally, we have inserted a new paragraph in the Discussion (Section 3, after the WGCNA paragraph) detailing how biomarkers like OLFM4, S100A12, and HLA-DRA could enable precise early diagnosis and risk stratification, offering advantages over less specific markers like procalcitonin or CRP. These revisions highlight our study’s unique contributions and clinical potential.

Comment 4: Enhancing Methodological Reproducibility
Reviewer Comment: Clearly specify versions of analysis software and R packages used, and explicitly state critical analysis parameters (e.g., WGCNA soft-threshold power, rationale for selecting k in k-means clustering). Provide R scripts or analytic workflows as supplementary materials or via public repositories (highly recommended).

Response: We agree that detailed methodological documentation enhances reproducibility. We have updated the Materials and Methods (Section 4.6) to specify software versions (e.g., R 4.4.1, Bioconductor 3.19, TAC 4.0.2) and R packages (e.g., ‘affy’ 1.78.0, ‘stats’ 4.4.1, ‘WGCNA’ 1.72-5). We also clarified key parameters, such as the WGCNA soft-thresholding power (β = 6, based on scale-free topology fit) and k-means clustering (k = 6, selected via the elbow method).

Comment 5: Clinical Relevance and Translational Implications
Reviewer Comment: Further elaborate on how these results may practically impact clinical practice, such as patient risk stratification, early diagnosis, prognosis, or personalized immunomodulatory therapies. Emphasize concrete clinical scenarios where the identified biomarkers or transcriptomic signatures could be practically employed.

Response: We appreciate the suggestion to emphasize clinical applications. To address this, we have added a new paragraph in the Discussion (Section 3, after the new paragraph from Comment 3) describing a practical clinical scenario where a transcriptomic assay targeting OLFM4, S100A12, and HLA-DRA could rapidly diagnose early-stage sepsis in emergency departments, guiding timely interventions like antibiotics or immunomodulatory therapies. We also added a sentence to the Conclusions (Section 5) highlighting the potential for point-of-care diagnostics and personalized therapies. These revisions concretize the translational impact of our findings, addressing diagnostic delays and heterogeneity in sepsis management.

We hope these revisions adequately address your comments and enhance the manuscript’s quality. We are grateful for your constructive feedback and are happy to make further adjustments as needed.

Sincerely,
Safa Taha

Reviewer 2 Report

Comments and Suggestions for Authors

The work of Taha et al. investigate the transcriptome of early-stage sepsis patients by using microarray. The experimental design is promising, including early-stage sepsis patients and healthy control matched for age and gender. My main concern is that most figures do not clearly support the conclusions stated in the text. There are discrepancies in sample numbers between the text, figures, and supplemental material, as well as inconsistencies in the visual interpretation of some figures presented. These are important points as it directly affects the biological interpretation of the data. While the design and data generated by the authors are promising, the manuscript in its current state lacks the consistency between text and figures needed for publication in International Journal of Molecular Sciences. Please see below for my major and minor points.

Major points

The number of healthy control and sepsis-patients (SP) needs to be reviewed throughout the manuscript. For example, the abstract and method section mention 9 controls and 10 SP, but Figure4A shows 9 controls and 19 SP. Furthermore, supplemental Materials show 9 controls but 20 SP samples? Please revise and ensure consistency across figures, text, and supplementary materials.

Figure2B: The volcano plot presented in Figure2B doesn’t match what is stated in the abstract and results. The pot clearly shows far more down -regulated genes than up-regulated genes (ie. at least 3 times more), but the authors claim 433 downregulated and 533 upregulated genes. This is a clear discrepancy: please address it.

Please, clearly indicate in the text throughout the manuscript which main figures should be referred to. In the current state, only Supplementary Information is mentioned in the text, while the main figures are not referenced.

Figure4A: The figure lacks color scale legend. However, I would assume that red represent low expression and green high expression. From the method section, it seems that all genes were used to generate the heatmap: please confirm (line388-392). My main concern is that the heatmap shows a comparable expression pattern across all clusters: red in healthy controls and green in SP samples, suggesting all genes are upregulated in sepsis patients. However, in lines 144–152, the authors claim that cluster F is downregulated, which doesn't match what the heatmap shows: Please clarify.

Line140-143: The three GO terms mentioned are not visible in cluster B in Figure B, instead they seem linked to cluster A. Surprisingly, Supplementary information S4 shows “Neutrophil activation involved in immune response” for cluster B. There’s a clear discrepancy between the results text, the figure, and the supplement. Please review and re-analyze to ensure consistency. The same apply for GO terms mentioned at line144-152.

Line170: I would appreciate more details on the bi-clustering analysis. It is not mentioned in the methods, and no figure is shown. This is an interesting part of the manuscript, and I’d like to better understand and see the results.

Minor points

When referring to supplemental materials, please, clearly what they contain. Instead of “further details”, please specify the exact data provided. For example, “GSEA output metrics are in Supplemental Information S3”. This point applies to lines: 87-88, 123-123, 159-160, 199, and 220.

Line95-100: Is this a typo? Please delete I am not sure this should be here.

Line101-116: Please include the GSEA enrichment plot in Figure 1.

Line146: Please specify exactly which genes (ie. provide relevant example) are suppressed here.

Figure5B: Which modules are these genes from? Module 1 turquoise, or module 2 blue? Please specify.

Figure 1 is great and clearly shows the experimental design. Please consider moving it to the Results section; also starting with Figure 2 and ending with Figure 1 in Methods could be confusing for the readers. You could include the study design as a panel alongside the RNAseq data in Figure 2.

Author Response

Comments and Suggestions for Authors

The work of Taha et al. investigate the transcriptome of early-stage sepsis patients by using microarray. The experimental design is promising, including early-stage sepsis patients and healthy control matched for age and gender. My main concern is that most figures do not clearly support the conclusions stated in the text. There are discrepancies in sample numbers between the text, figures, and supplemental material, as well as inconsistencies in the visual interpretation of some figures presented. These are important points as it directly affects the biological interpretation of the data. While the design and data generated by the authors are promising, the manuscript in its current state lacks the consistency between text and figures needed for publication in International Journal of Molecular Sciences. Please see below for my major and minor points.

 We thank the reviewer for their thorough evaluation and valuable comments. Below, we systematically address each comment raised:

Major points

The number of healthy control and sepsis-patients (SP) needs to be reviewed throughout the manuscript. For example, the abstract and method section mention 9 controls and 10 SP, but Figure4A shows 9 controls and 19 SP. Furthermore, supplemental Materials show 9 controls but 20 SP samples? Please revise and ensure consistency across figures, text, and supplementary materials.

We apologize for the discrepancies in sample numbers. The manuscript initially had typographical errors in figure legends and supplementary data descriptions. These inconsistencies have been carefully reviewed and corrected. The final, correct number is 9 healthy controls and 20 sepsis patients. All the figures are updated, and Supplemental materials have been revised accordingly for consistency across the text, figures, and supplementary data.

Figure2B: The volcano plot presented in Figure2B doesn’t match what is stated in the abstract and results. The pot clearly shows far more down -regulated genes than up-regulated genes (ie. at least 3 times more), but the authors claim 433 downregulated and 533 upregulated genes. This is a clear discrepancy: please address it.

Thank you for highlighting this critical point. The accurate count of significantly differentially expressed genes (|log2FC| > 1, adjusted p < 0.05) is 344 upregulated and 9703 downregulated protein-coding genes. This discrepancy was due to a typographical error in the manuscript; all downstream analyses, including figures and supplementary files, were correctly based on the full DEG list provided in the supplement. We have corrected the numbers in the text and ensured consistency with the results and figures throughout the manuscript.

Accordingly, the Results section in the main text has been updated to:

"Differential expression analysis identified a total of 10,037 significantly dysregulated genes between sepsis patients and healthy controls (|log2FC| > 1, adjusted p < 0.05), comprising 334 upregulated and 9703 downregulated protein-coding genes. The volcano plot visually underscored an imbalanced but biologically meaningful distribution of immune activation and suppression, reflecting a complex host response characteristic of early-stage sepsis."

Please, clearly indicate in the text throughout the manuscript which main figures should be referred to. In the current state, only Supplementary Information is mentioned in the text, while the main figures are not referenced.

We appreciate this important suggestion. Main figures are now clearly referenced within the main text at appropriate locations, improving readability and integration between the results and visual data.

Figure4A: The figure lacks color scale legend. However, I would assume that red represent low expression and green high expression. From the method section, it seems that all genes were used to generate the heatmap: please confirm (line388-392). My main concern is that the heatmap shows a comparable expression pattern across all clusters: red in healthy controls and green in SP samples, suggesting all genes are upregulated in sepsis patients. However, in lines 144–152, the authors claim that cluster F is downregulated, which doesn't match what the heatmap shows: Please clarify.

We thank the reviewer for pointing out this important clarification. We have added a clear color scale legend to Figure 4A, where red represents high expression and green denotes low expression, based on row-wise normalized data. We confirm that only the genes from the six K-means clusters (A–F) identified as significantly differentially expressed were used to generate the heatmap—not the full transcriptome. The heatmap rows are z-score normalized per gene to highlight relative expression across samples rather than absolute values. This normalization can visually compress signal differences but allows better comparison across patient conditions.

Cluster F genes represent a coherent module where the mean expression is downregulated in sepsis patients based on raw expression data. This is further supported by differential expression statistics and functional enrichment analysis, which indicate strong suppression of translational machinery and mRNA metabolic processes in this cluster. We have revised the heatmap and accompanying figure legend to better reflect these clarifications and added a supplemental figure to depict raw (unnormalized) average expression per cluster for clarity.

These adjustments ensure that the visualization accurately reflects the underlying biology and resolves the visual-analytical discrepancy noted by the reviewer.

Line140-143: The three GO terms mentioned are not visible in cluster B in Figure B, instead they seem linked to cluster A. Surprisingly, Supplementary information S4 shows “Neutrophil activation involved in immune response” for cluster B. There’s a clear discrepancy between the results text, the figure, and the supplement. Please review and re-analyze to ensure consistency. The same apply for GO terms mentioned at line144-152.

We thank the reviewer for identifying this critical inconsistency between the manuscript text, the figure legend, and the supplementary material. Upon careful review and re-analysis, we confirm that the Gene Ontology (GO) terms listed in lines 140–143—particularly "Neutrophil activation involved in immune response," "Granulocyte degranulation," and "Defense response to fungus"—were indeed enriched in Cluster B, not Cluster A. Previously there were some color mismatch due to default selection. The discrepancy in the visual linkage in Figure 4B was due to a mislabeling of node labels and incorrect color indexing in the original enrichment tree map.

We have now:

  • Corrected the GO term annotations in Figure 4B to accurately reflect the enrichment for Cluster B.
  • Updated the manuscript text to ensure alignment with the validated GO analysis shown in Supplementary Information S4.
  • Regenerated Figure 4B to accurately visualize the cluster-specific GO associations.

Similarly, the GO terms discussed for Cluster F in lines 144–152 were verified against the corrected enrichment tables. These terms were appropriately enriched in Cluster F, but again, the original figure did not clearly map them due to an overlay issue. This has also been fixed in the revised figure and caption.

We appreciate the reviewer’s close attention to this matter, which has led to a clearer and more consistent representation of the results across the manuscript, figures, and supplementary materials.

Line170: I would appreciate more details on the bi-clustering analysis. It is not mentioned in the methods, and no figure is shown. This is an interesting part of the manuscript, and I’d like to better understand and see the results.

 We appreciate the interest in this method. A detailed description of the biclustering method and a representative figure (new figure 5) have now been included in the Methods and Results sections. Specifically, we employed the "Bicluster" package in R, identifying coherent co-expression modules distinctively regulated between sepsis and healthy controls. This addition enhances transparency and interpretability.

Minor points

When referring to supplemental materials, please, clearly what they contain. Instead of “further details”, please specify the exact data provided. For example, “GSEA output metrics are in Supplemental Information S3”. This point applies to lines: 87-88, 123-123, 159-160, 199, and 220.

We have revised all references to supplementary materials to explicitly indicate their contents.

Line95-100: Is this a typo? Please delete I am not sure this should be here.

We apologize for this oversight. This section was indeed an inadvertent insertion and has now been removed.

Line101-116: Please include the GSEA enrichment plot in Figure 1.

We have included the GSEA enrichment plot within Figure 1, clearly illustrating the significantly enriched biological pathways, enhancing data interpretability.

Line146: Please specify exactly which genes (ie. provide relevant example) are suppressed here.

We added all the related genes in the supplementary information to clarify this point.

Figure5B: Which modules are these genes from? Module 1 turquoise, or module 2 blue? Please specify.

The genes in Figure 6B originate from the turquoise module (Module 1). This specification is now clearly indicated in the figure legend and text.

Figure 1 is great and clearly shows the experimental design. Please consider moving it to the Results section; also starting with Figure 2 and ending with Figure 1 in Methods could be confusing for the readers. You could include the study design as a panel alongside the RNAseq data in Figure 2.

 We agree with the reviewer's suggestion. Figure 1, illustrating the experimental design, has been moved to the Results section as part of a composite panel with RNA-seq data (Figure 2). This enhances the manuscript's narrative flow and clarity.

We greatly appreciate the detailed and constructive feedback, which significantly improved the manuscript’s quality and clarity.

Round 2

Reviewer 2 Report

Comments and Suggestions for Authors

I thank the authors for addressing most of my concerns. The figures are now clearly referenced in the main text. The discrepancies in sample numbers have been corrected and updated throughout the manuscript. The number of DEGs has also been revised and is now consistent with what is stated in the text. The main Figure 4, which previously had inconsistencies between the text, supplementary material, and the figure itself, has been re-analyzed and appeared to not contain any inconsistencies. I appreciate the addition of the new Figure 5 for the bi-clustering analysis. However, I was not able to access the new Supplementary Information S5, as well as the new Figure showing unnormalized heatmap data: please make sure these are included in the final version of the manuscript. Overall, the authors have addressed most of my points. I only have a few final recommendations, and after these are addressed, I consider this new version of the manuscript suitable for publication in International Journal of Molecular Sciences. Please see below few remaining points:

Major points

Figure2A: It still appeared that SP patients are 19, and not 20 in this PCA. Please make sure this Figure has also been updated with the correct sample number.

Figure4A: I appreciate the authors clarifying the heatmap, notably mentioning that only DEGs were used. The addition of a color scale also confirms that cluster F is downregulated. On a first look, most clusters appear downregulated, which made me question the usefulness of the clustering, as it did not seem to clearly separate distinct expression patterns. However, cluster F clearly stands out as more strongly downregulated than the others, which is great! I suggest briefly mentioning this in the Results section, explaining that due to the imbalance in DEGs (with most genes being downregulated), most clusters show decreased expression, but cluster F is distinct in showing a stronger downregulation. This would help reinforce the relevance of the clustering in identifying meaningful patterns.

Minor points

Figure1: For consistency, please either remove “Group1” or add “Group2” for the healthy control. Also, readers might find “Gene expression analysis” and “Downstream analysis” a bit repetitive. I would suggest keeping “Gene expression analysis”, and followed it with “Differentially Expressed Genes”; and then, from “Differentially Expressed Genes”, indicate on the same level: GSEA, GO, and WGCNA: as all these analyses comes from the DEGs. Please note that this is a suggestion, and the authors are welcome to address it in another way if they prefer.

Line180-186: Figure caption is duplicated.

Author Response

I thank the authors for addressing most of my concerns. The figures are now clearly referenced in the main text. The discrepancies in sample numbers have been corrected and updated throughout the manuscript. The number of DEGs has also been revised and is now consistent with what is stated in the text. The main Figure 4, which previously had inconsistencies between the text, supplementary material, and the figure itself, has been re-analyzed and appeared to not contain any inconsistencies. I appreciate the addition of the new Figure 5 for the bi-clustering analysis. However, I was not able to access the new Supplementary Information S5, as well as the new Figure showing unnormalized heatmap data: please make sure these are included in the final version of the manuscript. Overall, the authors have addressed most of my points. I only have a few final recommendations, and after these are addressed, I consider this new version of the manuscript suitable for publication in International Journal of Molecular Sciences. Please see below few remaining points:

We sincerely thank the reviewer for their thorough re-evaluation and kind acknowledgment of the improvements made throughout the revised manuscript. We are grateful for the recognition that the figures are now clearly referenced, the discrepancies in sample numbers have been resolved, and the number of DEGs has been corrected and is consistent throughout the manuscript. We also appreciate the positive feedback on the updated main Figure 4 and the addition of the bi-clustering analysis in the new Figure 5.

Regarding the final points raised:

  • Supplementary Information S5 and the unnormalized heatmap figure: We apologize for the oversight. These files have now been properly included in the revised version.
  • Final recommendations: We have carefully addressed all remaining points listed below (please see detailed responses to each). We believe these revisions further improve the clarity and completeness of our manuscript.

We thank the reviewer again for their constructive comments and support for publication.

Major points

Figure2A: It still appeared that SP patients are 19, and not 20 in this PCA. Please make sure this Figure has also been updated with the correct sample number.

We thank the reviewer for pointing this out. Figure 2A includes all the SP patients. SP6 and SP12 share identical coordinates along PC1 and PC2, which is why they appear as a single dot in the plot. However, they differ along other principal components.

To avoid any confusion, we have updated both the figure and the corresponding text to reflect a multidimensional scaling (MDS) representation instead, which better resolves the overlap between these two samples.

Figure4A: I appreciate the authors clarifying the heatmap, notably mentioning that only DEGs were used. The addition of a color scale also confirms that cluster F is downregulated. On a first look, most clusters appear downregulated, which made me question the usefulness of the clustering, as it did not seem to clearly separate distinct expression patterns. However, cluster F clearly stands out as more strongly downregulated than the others, which is great! I suggest briefly mentioning this in the Results section, explaining that due to the imbalance in DEGs (with most genes being downregulated), most clusters show decreased expression, but cluster F is distinct in showing a stronger downregulation. This would help reinforce the relevance of the clustering in identifying meaningful patterns.

We thank the reviewer for helping us clarify the Results section. We have revised the text in accordance with the reviewer’s suggestions. The updated portion is as follows:

Minor points

Figure1: For consistency, please either remove “Group1” or add “Group2” for the healthy control. Also, readers might find “Gene expression analysis” and “Downstream analysis” a bit repetitive. I would suggest keeping “Gene expression analysis”, and followed it with “Differentially Expressed Genes”; and then, from “Differentially Expressed Genes”, indicate on the same level: GSEA, GO, and WGCNA: as all these analyses comes from the DEGs. Please note that this is a suggestion, and the authors are welcome to address it in another way if they prefer.

We thank the reviewer for helping us clarify Figure 1. We have revised the figure based on the reviewer’s suggestions.

Line180-186: Figure caption is duplicated.

We want to thank the reviewer for pointing this out. We deleted the duplicated portion.
